# Fitting Into Any Shape: A Flexible LLM-Based Re-Ranker With Configurable Depth and Width

## Abstract

Large language models (LLMs) provide powerful foundations to perform fine-grained text re-ranking. However, they are often prohibitive in reality due to constraints on computation bandwidth. In this work, we propose a **flexible** architecture called **Matroyshka Re-Ranker**, which is designed to facilitate **runtime customization** of model layers and sequence lengths at each layer based on users' configurations. Consequently, the LLM-based re-rankers can be made applicable across various real-world situations.

The increased flexibility may come at the cost of precision loss. To address this problem, we introduce a suite of techniques to optimize the performance. First, we propose **cascaded self-distillation**, where each sub-architecture learns to preserve a precise re-ranking performance from its super components, whose predictions can be exploited as smooth and informative teacher signals. Second, we design a **factorized compensation mechanism**, where two collaborative Low-Rank Adaptation modules, vertical and horizontal, are jointly employed to compensate for the precision loss resulted from arbitrary combinations of layer and sequence compression.

We perform comprehensive experiments based on the passage and document retrieval datasets from MSMARCO, along with all public datasets from BEIR benchmark. In our experiments, Matryoshka Re-Ranker substantially outperforms the existing methods, while effectively preserving its superior performance across various forms of compression and different application scenarios. Our source code has been uploaded to *this anonymous repository*.

## Keywords

Text Retrieval, LLM-based Re-rankers, Lightweighting, Flexibility

## 1 Introduction

Text retrieval is crucial for many real-world applications, like web search, question answering, and retrieval-augmented generation [2, 19, 20, 31]. To retrieve relevant documents from a vast database, text retrieval typically employs a multi-stage process. Initially, it uses a combination of hybrid first-stage retrieval methods, such as embedding models [3, 18, 30, 53] and sparse retrievers [6, 26, 35, 41], to gather a comprehensive set of candidate documents for the query. Subsequently, a re-ranking model performs a fine-grained selection to identify the most relevant documents. Although the re-ranking

*Conference'17, July 2017, Washington, DC, USA*
© 2024 Copyright held by the owner/author(s). Publication rights licensed to ACM.
ACM ISBN 978-x-xxxx-xxxx-x/YY/MM
https://doi.org/10.1145/nnnnnnn.nnnnnnn

step is applied only to the candidates, it determines the final document order and thus significantly impacts the retrieval quality. Compared to first-stage retrieval methods, re-ranking models are computationally expensive but more precise in assessing the relevance between query and document. In the last few years, cross-encoders built on top of pre-trained models, e.g., monoBERT [34], monoT5 [34], and rank-T5 [57], have been widely used for this purpose. Meanwhile, same-period research shows that the re-ranking performance can be improved consistently with the expanded size of cross-encoders [55, 57]. Consequently, large language models (e.g., GPT, Llama, Mistral) are further leveraged as the backbone for a new generation of re-rankers [28, 37, 38, 42], leading to state-of-the-art performance across various text retrieval benchmarks.

### 1.1 Existing Challenges

Despite higher precision, LLM-based re-rankers come with much larger computational costs compared to conventional methods. Notably, their time latency can be prohibitive for many real-time applications, and their memory demands may exceed the GPU capacity in production environments. Therefore, it is imperative to slim down these models properly before deploying them in practice. In general, lightweight LLM-based re-rankers can be realized in two ways. One is to directly finetune a smaller LLM, such as Phi-3 (3.8B) [1], as the re-ranking model. While straightforward, this approach is limited by the size of the available LLMs. The other one is to prune a small sub-structure of a customized size from a larger LLM, e.g., Llama-3 (70B) [9], and fine-tune the pruned backbone for re-ranking model [27, 52]. However, this ad-hoc pruning and finetuning is limited to one-time use. When handling different application scenarios, the pruning-and-finetuning operations may need to repeat, which leads to significant training overhead.

### 1.2 Our contributions

*1.2.1 Flexible Architecture.* We propose an novel architecture called **Matroyshka Re-Ranker**[1], which features for its **flexibility** and **runtime adaptability** (Figure 1). It is built on top of a full-scale LLM, which offers the highest re-ranking precision but is computationally intensive. Meanwhile, it enables lightening the full-scale model based on user configuration. In particular, it allows users to specify their needed depth $n$ and width $w_i$. *It then extracts the first-n layers of the model and compressed the i-th layer's length to $w_i$ based on each token's estimated importance in re-ranking*. With such an architecture, users can flexibly customize their re-rankers for the optimal cost-effectiveness. In particular, users can begin with the full-scale re-ranker, gradually reduce the model size along both the height and width dimensions, while continuously measuring re-ranking quality on the test set. Ultimately, they will arrive at the lightest model that maintains acceptable re-ranking performance.

---

[1]Matryoshka, or stacking dolls, nesting dolls, et al., are dolls of decreasing size placed one inside another. It's used to describe the flexible nature of our model architecture.

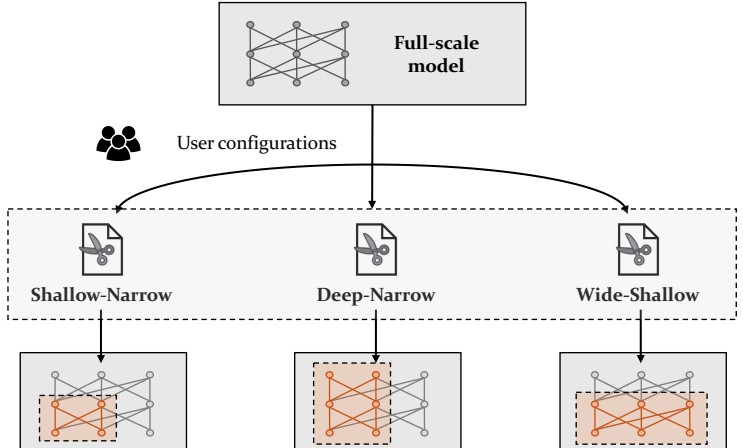 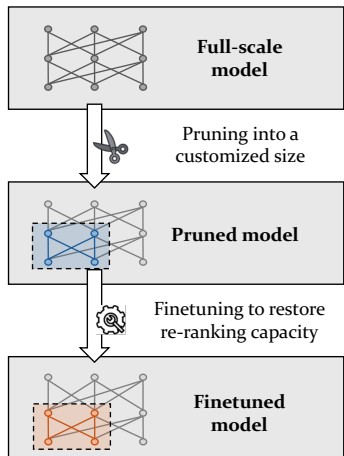

**A. Matryoshka Re-ranker**

**B. Ad-hoc Prune-Finetune**

**Figure 1: Matryoshka re-ranker (A) can be directly customized into arbitrary shapes based on users' configurations. In contrast, the conventional method (B) needs to perform ad-hoc pruning of the full-scale model and fine-tune it for each specific scenario.**

*1.2.2 Optimized Performance.* While the increased flexibility facilitates people's usage, it may lead to a sub-optimal precision compared to the specially pruned-and-finetuned models. To address this problem, we innovate both *training* and *post-training* techniques to optimize Matryoshka Re-Ranker's performance.

• **Cascaded self-distillation**. In Matryoshka Re-ranker, the full-scale model delivers the highest precision, while each sub-structure is dominated by its super networks (i.e., those with more layers and greater lengths) in re-ranking capacity. The full-scale model's re-ranking performance is expected to be preserved by arbitrary lightweight models. Therefore, we innovate the training method as cascaded self-distillation. Starting with the full-scale model, we iteratively sample a series of sub-structures and make each sampled sub-structure learn to preserve its super networks' outputs via knowledge distillation. Since a model's outputs can provide informative and smooth training signals for its sub-structures, this approach facilitates the fine-grained training of the Matryoshka Re-Ranker. Besides, there is no need to introduce any external teachers, and all teacher scores can be computed in one feed-forward pass, which enables the training process to be time-efficient.

• **Factorized compensation mechanism**. We explore post-training to further compensate for remaining loss. Traditionally, a pruned or quantized model can restore its performance by learning a specialized LoRA compensator [7, 22, 27]. However, this approach is impractical for Matryoshka Re-ranker as it is prohibitive to train a specialized compensator for every possible sub-structure. To address this, we design a factorized compensation mechanism, featuring its collaborative LoRA compensators. It introduces two groups of LoRA modules: V-LoRA and H-LoRA, which are used to compensate for losses due to depth and width compression, respectively. Meanwhile, the compensator for an arbitrary sub-structure is created as the linear addition of corresponding modules from V-LoRA and H-LoRA. In this way, LoRA compensation is made available for Matryoshka Re-ranker at a feasible training expanse.

We conduct comprehensive experiments using the passage and doc retrieval tasks from MSMARCO [31], along with the 14 public

datasets from BEIR [43]. In our experiments, Matryoshka Re-ranker effectively preserves superior precision across various light-weight structures and application scenarios. Meanwhile, it significantly outperforms the existing public re-ranking models, leading to state-of-the-art performances on corresponding benchmarks. Our model and source code will be publicly released, which can facilitate both direct usage and distillation of embedding models.

In summary, our **contributions are threefold**: 1) the proposal of Matryoshka Re-Ranker, which is the first re-ranking model to support flexible depth and width customization at runtime; 2) the design of cascaded self-distillation and factorized compensation, which effectively optimize the performance; 3) the empirically verified effectiveness and the value as a broadly beneficial resource.

## 2 Related Work

### 2.1 Pre-trained Models For Text Retrieval

Pre-trained language models have been widely applied for text retrieval [13, 55]. Based on the way of how query and document are interacted, the applications can be partitioned into two paradigms: bi-encoder and cross-encoder. The bi-encoder is to represent query and document independently. Therefore, it can well-support the first-stage retrieval, which calls for time-efficient processing. One typical application form of bi-encoder is dense retrieval, where relevant documents to the query can be identified based on their embedding similarities [18, 40]. In recent years, numerous text embedding models have been continually developed by the community, e.g., Contriever [15], GTR [33], E5 [48, 50], BGE [3, 53], and OpenAI text embedding [30]. Such embedding models have substantially improved the precision and generality of dense retrieval, making it a major retrieval strategy for real-world applications. Besides, the pre-trained language models can also be used to estimate the term importance [6, 10, 24], which contributes to the precision of lexical retrieval. Different from bi-encoder, the cross-encoder seek to establish the in-depth interaction between query and document, which facilitates fine-grained modeling of relevance. As such, it is widely

applied for the re-ranking step. In this paradigm, the pre-trained models can be directly fine-tuned to regress the classification logit of re-ranking [36]. Meanwhile, it can also leverage the generation likelihood on top of sequence-to-sequence learning [34, 57], which leads to more flexible computation of re-ranking scores.

While substantial progress was made by preliminary pre-trained models like BERT [8] and T5 [39], previous research indicated that text retrieval quality can be consistently improved when the model size continues to expand [30, 33, 57]. Following this empirical principle, people start make active use of large language models (LLMs) for text retrieval applications, and such a trend becomes significantly pronounced after the popularity of ChatGPT. Thanks to their instruction-following capabilities, LLMs can be directly prompted to perform various text retrieval tasks, such as re-ranking [37, 42] and query expansion [11, 51]. Additionally, LLMs can be fine-tuned specifically for text retrieval, leading to even better performance with smaller model sizes [21, 28, 38, 56]. The application of LLMs has led to significant improvements in text retrieval quality across many popular evaluation benchmarks. Notably, the top performers on MTEB [29] are all powered by LLM backbones. Furthermore, the re-ranking performances on MSMARCO [31] and BEIR benchmark [43] have also been dramatically improved by fine-tuned LLMs, such as RankLlama [28] and RankGemma [53].

## 2.2 Lightweight Re-Rankers

While using large models can enhance the precision of text retrieval, it also incurs substantial computational costs, which are prohibitive for many real-world applications. Traditionally, LLM-based re-rankers can be accelerated from two directions. One is to directly finetune smaller-scale LLM backbones. Recently, a number of powerful lightweight LLMs were developed by different organizations, such as the Phi-series LLMs from Microsoft [1], Llama-3-series LLMs from Meta [9], and Qwen-2-series from Alibaba [54]. However, this approach is still constrained by the size and architecture of the available models. The other one is to rely on ad-hoc pruning and finetuning [27, 52]. In particular, people can either prune an initial LLM backbone into their needed architectures and finetune the pruned model for re-ranking; alternatively, they can also prune a well-trained re-ranker and then restore its re-ranking capacity via continual fine-tuning. While this method allows for customized lightweight re-rankers, each generated model can only be used for a single specific task. When a new user requirement is presented, the pruning and fine-tuning processes must be repeated, which leads to significant training costs due to this limitation. In contrast, our proposed method allows for the production of customized lightweight re-rankers at runtime, i.e., after the full-scale model has been well trained and deployed for service, which eliminates the need for continual fine-tuning. To the best of our knowledge, this is the first text re-ranker of its kind, providing significant convenience for practical applications.

## 3 Matryoshka Re-Ranker

### 3.1 Preliminaries

The re-ranking model is used to perform fine-grained analysis for the relevance between query $q$ and candidate documents $\{d_1, ..., d_n\}$. Following the typical setting of pointwise learning-to-rank, the

model will generate an explicit re-ranking score for each query-doc pair, denoted as $\sigma(q, d)$. The re-ranking score is expected to precisely reflect the relevance, i.e. $\sigma(q, d_i) > \sigma(q, d_j)$ if $q$ is more relevant with $d_i$ than $d_j$. With superior semantic representation capabilities, the LLMs are applied in the form of cross-encoder for the re-ranking operation. Under this architecture, the query $q$ and document $d$ are concatenated as the following input template:

$$\text{Input} \leftarrow \text{"A:}\{q\}. \text{ B:}\{d\}.\{prompt\}\text{"}. \tag{1}$$

The above input is processed by LLM, e.g., Llama [44, 45], using the prompt "*Predict whether passage B contains an answer to query A?*". The last hidden state is linearly projected by the decoding head to predict the logit of "*Yes*", which is used as the re-ranking score:

$$\sigma(q, d) \leftarrow \text{Head}(\text{LLM}(\text{Input}).last\_hidden\_state)[\text{"Yes"}] \tag{2}$$

The re-ranking model is typically trained through contrastive learning [36, 57], where the discrimination likelihood is optimized for the ground-truth document $d^*$ (given all candidate docs: $D$):

$$\min. -\log \frac{\exp(\sigma(q, d^*)/\tau)}{\sum_{d \in D} \exp(\sigma(q, d)/\tau)}. \tag{3}$$

### 3.2 Flexible Architecture

The running cost of a re-ranker can be adjusted from two perspectives: 1) model's depth, 2) sequence length. Given a full-scale re-ranker, we can obtain a customized lightweight model by either removing the top layers from the full-scale model (depth customization), or gradually compressing the sequence at each layer (width customization). In this work, we propose Matryoshka re-ranker, which enables flexible customization from both perspectives.

*3.2.1 Depth customization.* Suppose the full-scale re-ranker is based on a LLM of N transformer layers, denoted as $\text{LLM}_{1,...,N}$. Unlike the traditional methods where the re-ranking score can only be computed from the last layer, we propose to learn the following depth-adaptive architecture, which enables the re-ranking score to be computed based on the intermediate hidden-states of each layer:

$$\mathbf{H}_i \leftarrow \text{LLM}_{\leq i}(\text{Input}).hidden\_states, \tag{4}$$

$$\sigma_i(q, d) \leftarrow \text{Head}_i(\mathbf{H}_i[-1])[\text{"Yes"}]. \tag{5}$$

In this place, $\text{LLM}_{\leq i}$ is the first $i$ layers of the LLM and $\mathbf{H}_i$ is the hidden-states at the $i$-th layer. A layerwise decoding head $\text{Head}_i$ is introduced, which transforms $\mathbf{H}_i[-1]$, the last hidden state of the input, into the logit of "*Yes*" as the re-ranking score $\sigma_i(q, d)$.

*3.2.2 Width customization.* Suppose the $i$-th layer of the re-ranker produces a sequence of hidden states of length $L$: $\mathbf{H}_i[1], ..., \mathbf{H}_i[L]$. Instead of passing these hidden states directly to the next layer, they are compressed using weighted average pooling. Given an aggregation factor $k$ (an integer greater than 1), the $L$ hidden states in the $i$-th layer are grouped into consecutive intervals as follows:

$$\underbrace{\mathbf{H}_i[1] ... \mathbf{H}_i[k]}_{\text{group 1}}, \underbrace{\mathbf{H}_i[k+1] ... \mathbf{H}_i[2k]}_{\text{group 2}}, \underbrace{\mathbf{H}_i[L-k] ... \mathbf{H}_i[L]}_{\text{group } L/k}. \tag{6}$$

Considering that different hidden-states have distinct impacts to the re-ranking result, we perform *importance-aware merging* to

compress the sequence. For each interval, the included hidden states are merged through the following pooling operation:

$$\mathbf{H}'_i[j] \leftarrow \sum_{l=1\ldots k} \alpha_l * \mathbf{H}_i[j+l], \text{ where } \sum_{l=1\ldots k} \alpha_l = 1. \quad (7)$$

Knowing that the re-ranking score is computed based on $\mathbf{H}_i[-1]$, we can use the attention weight with the last token as an indicator of importance for each hidden state. Therefore, we can derive the pooling weight with the following computation:

$$\alpha_l \leftarrow \frac{\exp(a_{-1,j+l})}{\sum_{l=1\ldots k} \exp(a_{-1,j+l})}, \quad (8)$$

where $a_{-1,j+l}$ stands for the attention weight of the last token towards $\mathbf{H}_i[j+l]$ (using the average weight of all attention heads). Note that the above pooling operation can be selectively applied to a subset of the intervals, particularly those with the lowest combined attention weights. Thus, it allows for compressing the sequence into an arbitrary length based on user's requirement.

*3.2.3 User Configuration.* Users' may flexibly customize the architecture of their re-ranker by configuring the depth (the number of layers) and width (the sequence length of each layer) of LLM:

```
layer_1: sequence length L1,
layer_2: sequence length L2,
...
layer_n: sequence length Ln
```

Note that the total number of layers and the sequence length must not exceed the depth and width of the full-scale re-ranker: $n \leq N$, $L_i \leq L$. Additionally, the layerwise depth must be monotonically decreasing, that is, $L_i \leq L_{i+1}$. The compression factor at each layer is a hyper-parameter that can be determined by the user.

*3.2.4 Usage method.* The adjustment of the two dimensions, depth and width, will have varying effects on efficiency and precision, depending on the specific use case. For instance, reducing width may yield higher acceleration with minimal precision loss, while reducing depth may be more suitable for shorter inputs. In practice, Users can begin with the full-scale model, gradually reduce its size in both dimensions, and continuously measure the re-ranking precision on the validation dataset. Ultimately, the lightest sub-structure that maintains acceptable precision will offer the optimal trade-off between cost and effectiveness.

## 3.3 Cascaded Self-Distillation

While re-rankers can be directly trained by learning to discriminate the ground-truth documents from the first-stage candidates, the sub-structures within the full-scale model are likely to generate sub-optimal results due to the relatively lower capacity. To address this problem, we propose training the sub-structures to minimize precision loss relative to an upper-bound re-ranking model through knowledge distillation (KD) [23]. Typical KD methods involve training a specialized teacher and minimizing the prediction difference between teacher and student. Unfortunately, this paradigm is not well-suited for the Matryoshka re-ranker because the performance gap between the teacher and the student must be properly controlled: a too small gap provides little meaningful guidance, while

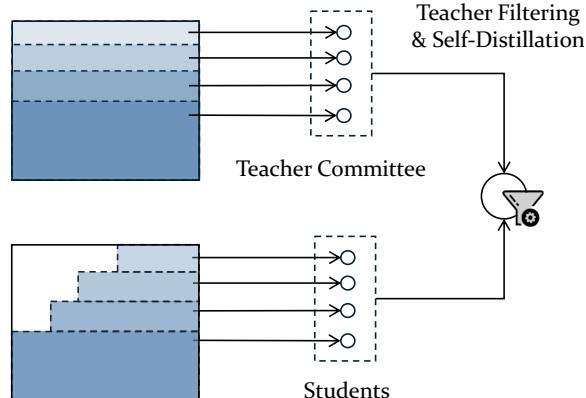

**Figure 2: Cascaded Self-Distillation. Upper: full-width layer-wise predictions are used as the teacher committee. Lower: students make selective use of teachers to distill knowledge.**

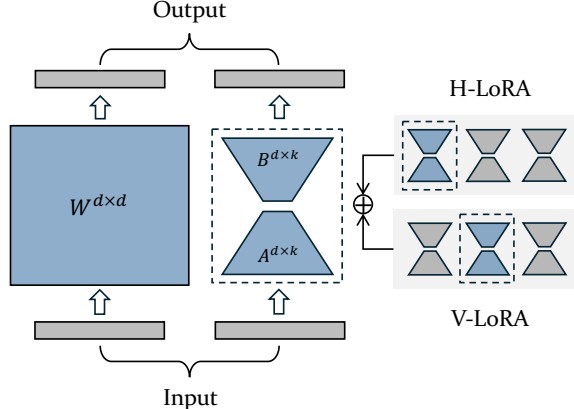

**Figure 3: Factorized compensation mechanism. The vertical (V-LoRA) and horizontal (H-LoRA) compensation modules are selected and added up to make up the precision loss.**

a too large gap makes it difficult to mimic the teacher's predictions. Since the Matryoshka re-ranker needs to learn various lightweight re-rankers of highly different sizes, it is impossible to find a universally appropriate teacher, nor is it feasible to introduce a specialized teacher for each individual lightweight re-ranker.

In this work, we design a novel training method based on the unique architecture of Matryoshka re-ranker. Before presenting its workflow, we present the following property as prior knowledge.

THEOREM 3.1. *A sub-structure $\mathcal{N}$ of Matryoshka re-ranker is dominated by its super-architecture $\mathcal{N}'$ in re-ranking precision: $\sigma_{\mathcal{N}'} > \sigma_{\mathcal{N}}$.*

A super-architecture of $\mathcal{N}$ refers to other networks within the full-scale model that have more layers and a larger width at each layer. We use the $\mathbb{I}(\cdot)$ to illustrate this relationship: $\mathbb{I}(\mathcal{N}'|\mathcal{N}) = 1$ if $\mathcal{N}'$ is a close super-architecture of $\mathcal{N}$; otherwise, it is 0.

On top of the above property, we propose to leverage the predictions from Matryoshka re-ranker itself for knowledge distillation, called **Cascaded Self-Distillation** (Figure 2). Specifically, we employ the whole full-width sub-structures (w/o. width compression)

as teacher committee $\mathcal{T}$: $\{t = \text{LLM}_\theta | \theta : \text{pre-defined teacher layers}\}$. During training, we sample sub-structures from the LLM as the students $\mathcal{S}$: $\{s = \text{LLM}_\phi | \phi : \text{all layers with sampled widths}\}$. The students make selective use of the teachers by filtering their super architectures from the committee, where knowledge distillation is performed. The knowledge distillation process can be formulated by the following optimization problem:

$$\min. - \sum_{\mathcal{S}} \sum_{\mathcal{T}} \mathbb{I}(t|s) * \frac{e^{\sigma_t(q,d^*)}}{\sum_{d \in D} e^{\sigma_t(q,d)}} \log \frac{e^{\sigma_s(q,d^*)}}{\sum_{d \in D} e^{\sigma_s(q,d)}}. \quad (9)$$

In this place, $\sigma_s(q, d)$ and $\sigma_t(q, d)$ represent student's and teacher's predictions of the relevance between query and document, and $D$ refers to the whole candidate documents.

**Comments**. Cascaded self-distillation provides diverse teacher signals, facilitating *fine-grained* training of student models. Besides, the training is *time-efficient*. Since teachers are full-width sub-structures and the students are sampled in a step-like manner (shallower students have larger widths), the re-ranking scores can be computed in a single feed-forward pass for both teacher and student. Thus, the cost is equivalent to traditional KD methods, which involve one student and one specialized teacher model.

## 3.4 Factorized Compensation

The directly trained Matryoshka Re-Ranker is further improved by post-training. For pruned or quantized models from a full-scale LLM, it is common to perform continual PEFT fine-tuning [28, 52], where an extra LoRA module is employed to compensate the potential performance loss [7, 14]. Despite its straightforward nature, the tradition method is applied for the compensation of one specific model. Therefore, it is not unsuitable for Matryoshka Re-Ranker because compensations need to be made for arbitrary sub-structures extracted from the full-scale model.

In this work, we propose the factorized compensation mechanism, where two collaborative adapter lists: V-LoRA (vertical) and H-LoRA (horizontal), are employed (shown as Figure 3). Particularly, V-LoRA contains a list of LoRA adapters, each of which is corresponding to a specific layer. While H-LoRA is composed of another list of LoRA adapters, each one is corresponding to a specific width compression factor $k$ (see Eq. 6):

$$\text{V-LoRA} : \{\theta_i^v | i = 1, ..., N\}, \text{ H-LoRA} : \{\theta_k^h | k = 2, ..., M\} \quad (10)$$

For an arbitrary sub-structure (with layerwise projection matrix denoted as $\{W_l\}_L$), the LoRA adapter is created as the linear addition of corresponding V-LoRA and H-LoRA adapters: $\theta_l \leftarrow \theta_l^v + \theta_k^h$, where $k$ is the compression factor at the $l$-th layer. As a result, layerwise projection matrix is updated as: $W_l \leftarrow W_l + \theta_l.A^T\theta_l.B$ ($A$ and $B$ stand for the low-rank matrices of $\theta_l$ [14]).

Compared to the standard LoRA adapter of a full-scale LLM, the proposed formulation introduces only H-LoRA as additional parameters. As a result, the compensation modules can be efficiently trained through continual fine-tuning. For each sampled module, the adapters are generated based on its substructure, followed by learning to optimize the re-ranking objective. Given that there are minimal number of new parameters and the backbone LLM has been well-trained in prior stage, the training process can converge quickly to a competitive performance.

## 4 Experiments

In this section, empirical studies are performed to explore the following research questions regarding the effectiveness and efficiency of Matryoshka re-ranker. **RQ 1.** Whether it can achieve high-quality performances for lightweight sub-structures extracted from the full-scale re-ranker. **RQ 2.** Can it maintain strong performances across diverse evaluation scenarios. **RQ 3.** Can it flexibly support various forms of compression? **RQ 4.** How do different working conditions and technical factors influence the re-ranker's performance.

## 4.1 Settings

*4.1.1 Datasets.* Matryoshka re-rankers are trained respectively with two datasets from **MSMARCO** [32]: 1) passage, and 2) document. Specifically, MSMARCO-passage contains 500,000 training queries, 6,980 dev queries, and a corpus of 8.8 million passages; while MARCO-document contains 300,000 training queries, 5,193 dev queries, and a corpus of 3.2 million documents.

Following the previous studies, the fine-tuned re-rankers are evaluated for their passage and document retrieval performance leveraging the dev queries provided by MSMARCO. Besides, the models are evaluated based on the testing queries released by **DL'19** [5] and **DL'20** [4]. Finally, Matryoshka re-ranker is also evaluated for its general text-retrieval performance using **BEIR** [43]. It is a miscellaneous benchmark of 18 text-retrieval datasets (the 14 public ones are used in our experiment), covering different types of domains (such as Wikipedia, bio-medical, finance, social-media) and tasks (e.g., passage retrieval, question retrieval, argument retrieval).

*4.1.2 Evaluations.* We focus on two types of methods in our evaluation. One is the **existing baseline re-rankers**, including 1) the classic *BERT-like re-rankers*, like MonoBERT [36], MonoT5 [34], and BGE [3, 53], where pre-trained models, e.g., BERT and T5, are fine-tuned as the re-rankers; 2) the *fine-tuned LLM-based re-rankers*, like RankLlama [28], RankVicuna [37], in which open-sourced LLMs are fine-tuned as the re-rankers; 3) the *prompted LLM-based re-rankers*, such as RankGPT-gpt-4 [42], where proprietary LLMs, like Chat-GPT, are prompted for re-ranking. The other one are the full-scale and specially pruned-and-finetuned re-rankers, which present the performance upperbound (called **specialized upperbound**).

The re-ranking is primarily made based on the top-100 candidates returned by BGE-EN-v1.5 large [53]. Additional analysis is also conducted with various first-stage retrievers and different numbers of first-stage candidates. The performance is measured by classic re-ranking metrics, like MRR [46] and NDCG [16], as required by each specific benchmark.

*4.1.3 Implementations.* Matryoshka re-ranker is primary trained based on the Mistral-7B model [17] which comprises 32 layers in total. Meanwhile, extended study is conducted with other popular LLM backbones, including Llama-3-8B [25] and Gemma-2-9B [12]. For passage ranking, the maximum input length is 224, the batch size is 128, the number of negative samples is 15; while for document ranking, the maximum input length is 2048, the batch size is 128, the number of negative samples is 7. The model is trained by LoRA [14] with a rank of 32 and an alpha of 64, while the learning rate is $1e^{-4}$. The training process undergoes one epoch using Cascaded Self-Distillation, followed by another epoch of Factorized Compensation.

| | Passage Re-Ranking | | | | Document Re-Ranking | | | |
|---|---|---|---|---|---|---|---|---|
| | Dev | | DL'19 | DL'20 | Dev | | DL'19 | DL'20 |
| Method | MRR@10 | NDCG@10 | NDCG@10 | NDCG@10 | MRR@100 | NDCG@100 | NDCG@10 | NDCG@10 |
| MonoBERT [36] | 38.02 | 44.82 | 68.61 | 67.70 | 35.14 | 43.31 | 65.50 | 62.29 |
| MonoT5-3B [34] | 38.99 | 47.00 | 71.29 | 70.42 | 37.07 | 44.90 | 66.90 | 67.75 |
| SimLM-Rank [47] | 43.41 | 49.93 | 73.34 | 72.67 | 40.38 | 51.18 | 65.70 | 63.74 |
| RankLLaMA [28] | 44.69 | 51.30 | 73.73 | **76.92** | 48.29 | 57.81 | 68.90 | 67.43 |
| RankVicuna [37] | - | - | 69.13 | 66.50 | - | - | 64.23 | 61.64 |
| RankGPT-gpt-3.5 [42] | - | - | 71.11 | 66.50 | - | - | 60.25 | 56.74 |
| RankGPT-gpt-4o [42] | - | - | 73.36 | 73.52 | - | - | 66.12 | 63.99 |
| Matryoshka lightweight | 44.85 | 51.50 | 74.65 | 75.45 | 49.66 | 58.97 | 70.40 | **68.10** |
| *Specialized upperbound (light)* | *44.86* | *51.53* | *74.24* | *75.54* | *49.64* | *58.95* | *70.48* | *67.35* |
| Matryoshka full-scale | **44.95** | **51.63** | **75.42** | 76.37 | **49.67** | **58.99** | **71.39** | 67.96 |
| *Specialized upperbound (full-scale)* | *44.93* | *51.64* | *74.97* | *75.67* | *49.70* | *59.19* | *69.77* | *67.53* |

Table 1: Re-ranking performances on MSMARCO. Specialized upperbounds are finetuned for light and full architectures.

## 4.2 Experiment Analysis

*4.2.1 MSMARCO Performance.* The passage and document retrieval performance on MSMARCO is shown in Table 1, where the top 100 candidates returned by BGE-EN-v1.5-large are re-ranked by all the included methods. There are two alternatives of our own approach. 1) **Matryoshka full-scale**, where the entire re-ranker is directly used without compression. 2) **Matryoshka lightweight**, in which the sequence length (i.e. width compression) is compressed by 50% at the 8th layer and the re-ranking score is computed from the 16-th layer's output (i.e. height compression). As such, it saves more than 60% of FLOPs with the lightweight model, and it saves more than 50% in inference time (achieving a 2× speedup compared to the full-scale model) with a sequence length of 1024.

In our experiments, the full-scale Matryoshka achieves the highest precision in the overall results, while the lightweight Matryoshka maintained similar performance at much lower costs. Notably, the relative gap between the lightweight and full-scale Matryoshka re-rankers is within 1‰ across most of the evaluation scenarios, despite that the computation cost has been reduced by more than 60%. This observation preliminarily indicates that strong cost-effectiveness of re-ranking can be realized by our proposed method, and subsequent experiments will further validate this point.

Additionally, it's noteworthy that both full-scale and lightweight models achieve very close performances compared with their specialized upperbounds, indicating that potential precision loss is effectively controlled while enhancing the model's flexibility. Our methods also exhibits notable advantages over the existing baseline, re-rankers, e.g., RankLlama, which leverages a same-szie LLM backbone, and RankGPT-gpt-4o, which is powered by a highly advanced proprietary LLM. The empirical advantage is even greater when compared to other popular re-rankers based on smaller models, such as MonoT5-3B and SimLM-Ranker. As a result, Matryoshka Re-Ranker can serve as a valuable resource for the community, offering precise and flexible re-ranking capabilities and acting as a superior teacher model for distilling other retrievers.

*4.2.2 BEIR Performance.* The retrieval performance on BEIR is shown in Table 2, where the included methods are used to re-rank the top-100 candidates returned BGE-EN-v1.5-large. Apart from the previous baselines on MSMARCO, we also introduce Jina-Rank-v2[2] and BGE Re-Ranker-M3[3] , which are popular general-purpose re-rankers trained from diverse datasets. We continue to employ two alternative configurations: Matryoshka lightweight (M lightweight) and full-scale (M full-scale) for comparison, following the same setting as the previous experiment.

According to the experimental results, our approach continues to demonstrate strong retrieval quality. Notably, both M lightweight and M full-scale can achieve significant advantages over a wide variety of competitive baselines, such as RankLlama, RankGPT (powered by GPT-4), and BGE Rank-M3 (one of the most widely applied re-rankers for general text retrieval tasks). The two methods also outperform the baselines in most individual scenarios, reflecting their strong generality. Besides, M lightweight maintains very close performance to M full-scale despite a reduction of over 60% in computation cost. In some specific tasks, M lightweight even slightly surpasses the full-scale alternative, which further highlights the effectiveness of our approach.

*4.2.3 Flexible Compression.* After the preliminary validation of effectiveness under the default form of compression, we continue to explore Matryoshka re-ranker's flexibility in supporting ad-hoc compression requirements.

We begin with the exploration of height compression, where the re-ranking score is computed at different layers of the model. Therefore, it will give rise to "wide-shallow" variations of the full-scale re-ranker. In our experiment, the re-ranker's height is gradually reduced from 32 to 6 in increments of 4 layers (i.e. 32, 28, 24 ... 8, 6), which results in growing reductions in FLOPs by 0%, 12.5%, 25%, ... 81.25% and leads to gradual reductions in inference time from 0%, 12%, 22%, to 78%. As illustrated in Figure 4 (red line), Matryoshka

---

[2]https://huggingface.co/jinaai/jina-reranker-v2-base-multilingual
[3]https://huggingface.co/BAAI/bge-reranker-v2-m3

| Model | MonoBERT | MonoT5-3B | SimLM | RankGPT* | RankLLaMA | BGE-Rank-M3 | Jina-Rank-v2 | M lightweight | M full-scale |
|---|---|---|---|---|---|---|---|---|---|
| NFCorpus | 36.16 | 40.91 | 33.91 | 38.47 | 29.36 | 34.85 | 37.73 | **41.96** | 41.50 |
| FIQA | 40.37 | 53.64 | 40.27 | - | 47.39 | 44.51 | 45.88 | 58.39 | **59.16** |
| SCIDOCS | 17.02 | 20.77 | 16.38 | - | 18.48 | 18.25 | 20.21 | 22.39 | **22.43** |
| FEVER | 83.48 | 85.91 | 84.13 | - | 86.30 | 90.15 | 92.44 | **94.79** | 94.76 |
| Arguana | 52.46 | 39.91 | 33.02 | - | 56.18 | 37.70 | 52.23 | 65.29 | **65.80** |
| Scifact | 72.37 | 76.88 | 68.00 | 74.95 | 72.17 | 73.08 | 76.93 | 79.87 | **80.25** |
| TREC-COVID | 76.51 | 82.52 | 77.98 | 85.51 | 84.26 | 83.39 | 80.89 | 84.88 | **85.54** |
| Climate-FEVER | 27.12 | 31.90 | 23.15 | - | 28.00 | 37.99 | 34.65 | 46.45 | **47.10** |
| HotpotQA | 75.27 | 77.99 | 74.81 | - | 78.83 | 84.51 | 81.81 | 87.80 | **87.87** |
| NQ | 59.15 | 65.66 | 60.74 | - | 66.13 | 69.37 | 67.35 | 74.69 | **75.08** |
| Quora | 72.84 | 83.75 | 53.03 | - | 85.58 | 89.13 | 87.81 | 90.74 | **91.02** |
| Touche | 26.96 | 29.14 | 38.01 | **38.57** | 36.73 | 33.22 | 32.45 | 30.20 | 30.96 |
| DBPedia | 45.33 | 49.45 | 46.15 | 47.12 | 48.74 | 48.15 | 49.31 | **52.55** | 52.30 |
| CQA | 37.84 | 45.74 | 33.65 | - | 39.46 | 38.24 | 40.21 | 48.02 | **48.46** |
| Average | 51.63 | 56.01 | 48.80 | - | 55.55 | 55.90 | 57.14 | 62.72 | **63.02** |

**Table 2: Text re-ranking performance on BEIR (measured by NDCG@10). The results marked with * are from reports, while the remaining results are reproduced by ourselves using the publicly released checkpoints.**

re-ranker stays robust to height compression: the impact on performance is almost ignorable until the model's height is reduced to 20, and it well-maintains its retrieval quality until the model's height is reduced to 12. Even only 6 layers left, the model remains effective, delivering substantial improvements over the first-stage retriever (as indicated by the purple dash line).

We also make exploration of width compression, where the re-ranking score is computed based on compressed input sequence. This approach results in "deep-narrow" variations of the full-scale re-ranker. We evaluate various compression settings, in which the sequence length is compressed at the 20th, 12th, and 4th layer respectively, meanwhile three compression factors: 2×, 4×, 8×, are applied (a factor $\alpha$ will compress the input to $1/\alpha$ of its original length). As such, it results in gradual reductions in FLOPs from 0%, 16%, 24%, to 71%, and leads to gradual reductions in inference time from 0%, 7%, 23%, to 73%. The experiment results, shown in Figure 4 (green line), demonstrate that Matryoshka re-ranker remains robust to width compression as well. Notably, the retrieval precision can be effectively preserved across different compression factors and starting layer of width compression.

We continue to explore more flexible forms by jointly applying height and width compression, in which the input sequence is compressed and the re-ranking score is computed at different layers. We still employ three factors: 2×, 4×, and 8×, meanwhile, width compression can repetitively take place all the way to the output layer. Therefore, it progressively reduces the FLOPs by 0%, 12.5%, 25% ... up to 80% and leads to gradual reductions in inference time by 0%, 12%, 22%, ... up to 74%. As illustrated by Figure 4 (blue line), the re-ranker's performance is still robust to the reduction of FLOPs, which is consistent with our previous observations in height and width-only scenarios. Additionally, this approach delivers even better preservation of retrieval performance compared to using height and width compression alone.

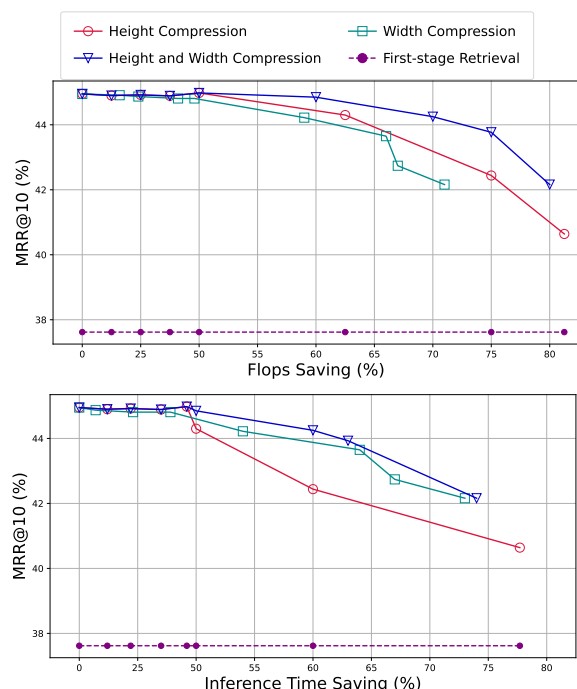

**Figure 4: Re-ranking performance (MRR@10) vs. FLOPs / inference time saving based on different forms of compression.**

The above results validate Matryoshka re-ranker's flexibility in supporting ad-hoc compression requirements. Such a flexibility enables users to best trade-off the retrieval quality and running cost in their individual application scenarios (as discussed in Sec 3.2.4).

*4.2.4 Ablation Studies.* The ablation studies are dedicated for two main purposes: 1) the exploration of Matryoshka re-ranker's effectiveness beyond its default setting in the main experiment, and

| Method | Size of 1st-stage candidates | | |
| --- | --- | --- | --- |
| | Top-50 | Top-100 | Top-200 |
| BM25 | 21.01 | 21.01 | 21.01 |
| BM25 + M-light | 39.08 | 40.56 | 41.89 |
| BM25 + M-full | 39.09 | 40.61 | 41.91 |
| BGE-small-en-v1.5 [53] | 36.08 | 36.08 | 36.08 |
| BGE-small-en-v1.5 + M-light | 44.44 | 44.82 | 44.94 |
| BGE-small-en-v1.5 + M-full | 44.45 | 44.87 | 45.04 |
| BGE-large-en-v1.5 [53] | 37.62 | 37.62 | 37.62 |
| BGE-large-en-v1.5 + M-light | 44.63 | 44.85 | 44.90 |
| BGE-large-en-v1.5 + M-full | 44.77 | 44.95 | 45.02 |
| E5-large-v2 [48] | 38.29 | 38.29 | 38.29 |
| E5-large-v2 + M-light | 44.73 | 44.88 | 44.93 |
| E5-large-v2 + M-full | 44.78 | 44.95 | 45.07 |
| E5-Mistral [49] | 37.67 | 37.67 | 37.67 |
| E5-Mistral + M-light | 44.74 | 44.91 | 44.99 |
| E5-Mistral + M-full | 44.91 | 45.07 | 45.15 |

**Table 3: Re-ranking performance (measured by MRR@10 on MSMARCO passage) based on diverse first-stage retrievers with different sizes of first-stage candidates (top 50, 100, 200).**

2) the analysis of each technical factor's impact in optimizing Matryoshka re-ranker's performance.

In the first place, we introduce a variety of first-stage retrievers of different types (embedding models and sparse method) and scales (small, large, and LLM-based), which include BM25, BGE-small, BGE-large, E5-large, and E5-Mistral. Additionally, we present different numbers of first-stage candidates for re-ranking, including the top-50, top-100, and top-200 candidates returned by various retrievers. The experimental results, shown in Table 3, demonstrate that both M-light and M-full significantly improve performance over all first-stage retrievers. Moreover, M-light fully maintains the performance of M-full, which is consistent with our observations in the previous experiments. As a result, these results indicate our effectiveness in dealing with diverse candidates of different quality quantity, and distributions.

Secondly, we employ three different LLM backbones to implement the re-ranking models: Mistral-7B [17], Llama-3-8B [25], and Gemma-2-9B [12]. Despite having similar scales, these LLM backbones take different model architectures and undergo varied pretraining, fine-tuning, and alignment processing. The experimental results, shown in Table 4, demonstrate competitive re-ranking performance across all implementations. Although Llama-3-8B and Gemma-2-9B are slightly larger than Mistral-7B, their performances are no better than the default option. This result can likely be attributed to the fact that the increased parameters for Llama-3-8B and Gemma-2-9B are mostly in the embedding layers and decoding heads: Llama-3-8B and Gemma-2-9B have 1.05B and 1.84B parameters, respectively, whereas Mistral-7B has only 0.26B. The increased token embeddings may benefit multi-lingual settings, but since our evaluation focuses on English, little benefit is observed.

| | Lightweight | | Full-scale | |
| --- | --- | --- | --- | --- |
| | MRR@10 | rel perf. | MRR@10 | rel perf. |
| Default | 44.85 | 99.7% | 44.95 | 100.3% |
| w/o Compensation | 44.35 | 93.0% | 44.55 | 94.8% |
| w/o Self-Distillation | 43.89 | 86.6% | 44.25 | 90.7% |
| Fist-stage retrieval | 37.62 | 0.0% | 37.62 | 0.0% |
| Specialized upperbound | 44.86 | 100.0% | 44.93 | 100.0% |
| Mistral-7B | 44.85 | – | 44.95 | – |
| Llama-3-8B | 44.52 | – | 44.82 | – |
| Gemma-2-9B | 44.76 | – | 44.91 | – |

**Table 4: Ablation studies based on MSMARCO-passage; rel perf. (%) measures the improvement over the first-stage retrieval baseline compared to the specialized upperbounds.**

Finally, we perform detailed analysis for each technical factor, with the following methods included for comparison. 1) Without Factorized Compensation, which eliminates post-training and only adopts cascaded self-distillation to fine-tune the model. 2) Without Self-Distillation, which learns Matryoshka Re-Ranker directly from the labeled data, instead of performing cascaded self-distillation.

The experiment results are shown in Table 4, where the relative performance (rel perf.) is measured based on the improvements over the first-stage retrieval baseline compared to the specialized upperbounds. As we can observe, the re-ranking precision is effectively preserved only when both techniques are jointly applied. Otherwise, the re-ranker' performances gradually decline with the removal of each technique, and this trend is more clearly highlighted from the relative performance perspective.

*4.2.5 Summary.* The following experiment insights can be made in response to our research questions.

- Along with its high flexibility, our approach achieves superior re-ranking results compared to the upperbound methods and popular baselines.
- Our method maintains strong performances across various benchmarks, with different sub-structure configurations, under diverse first-stage retrieval conditions, and using different backbone LLMs.
- Both optimization techniques make substantial contributions to the empirical performance.

## 5 Conclusion

In this paper, we introduce Matryoshka Re-Ranker, a method designed to facilitate the creation of lightweight LLM-based text re-ranking models. Our approach enables users to customize the re-ranker into arbitrary architectures by configuring its width and depth at runtime, which brings forth significant flexibility for people's usage. In addition, the model's performance is effectively optimized through cascaded self-distillation and factorized compensation during both training and post-training stages. Through comprehensive experimental analysis across various benchmarks and settings, Matryoshka Re-Ranker is verified as a precise and flexible method for real-world scenarios.

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
