# OpenReview forum: "Fitting Into Any Shape: A Flexible LLM-Based Re-Ranker With Configurable Depth and Width"
_ACM.org/TheWebConf/2025/Conference — WWW 2025 Poster_

### Official Review · Reviewer_Vs6G · 2024-11-11

**Novelty:** 2
**Technical Quality:** 2

**Review:**

The article presents a flexible architecture known as Matryoshka Re-Ranker, which is designed to accommodate different application scenarios by permitting runtime customization of model layers and the sequence length of each layer. Although this work has attained notable achievements in terms of flexibility and performance, in light of its content, I consider that there are several drawbacks as follows:


Low Innovation: The primary issue tackled in this paper is the significant consumption of time and space by large language models. However, the problems with existing fine-tuning and pruning work have not been meticulously analyzed. Instead, a so-called utilization of diverse application scenarios is directly put forward, yet the specific scenarios are not elucidated.


Usage Scenarios: Although the model is initially devised to adapt to a variety of application scenarios and maintain good performance under certain conditions, this paper fails to clearly enumerate the requirements of different scenarios and conduct experimental analyses in diverse situations. The experiments mentioned in the paper are mainly based on the MSMARCO and BEIR benchmark datasets, and the characteristics of these datasets might not fully represent all real-world application scenarios. Consequently, the performance of the model on other types of datasets still requires further verification.
﻿

Performance Loss: Although the paper mentions the introduction of a series of techniques to optimize performance, such as cascaded self-distillation and factor compensation mechanisms, these measures are mainly aimed at alleviating the precision loss resulting from model compression. This implies that when the model is compressed to a certain extent, there could still be a certain degree of performance degradation. The experimental results do not conduct a careful comparative analysis of the impact of introducing this technology on the model itself.

**Questions:**

1. What are the specific problems of the fine-tuning and pruning-based methods?

2. The paper mentions that performance is optimized through a series of techniques but there may still be performance degradation. So how to further improve these techniques or explore new methods to more effectively reduce the performance loss caused by model compression and more accurately evaluate its impact on the model?

3. The Matryoshka Re-Ranker adopts complex mechanisms that increase the system overhead. So how to simplify these mechanisms or reduce the overhead they bring without affecting the flexibility and performance of the model, thereby improving the overall efficiency of the model?

**Reviewer Confidence:**

4: The reviewer is certain that the evaluation is correct and very familiar with the relevant literature

**Scope:**

3: The work is somewhat relevant to the Web and to the track, and is of narrow interest to a sub-community

---

### Official Review · Reviewer_2h5u · 2024-11-29

**Novelty:** 5
**Technical Quality:** 4

**Review:**

This paper proposes a flexible architecture named Matryoshka Re-Ranker, which supports dynamic compression of LLMs by configurable depth and width while optimizing the performance of the compressed models through cascaded self-distillation and factorized compensation mechanisms. The research topic is meaningful, and the detailed evaluation is as follows:

Strengths:
1.	The research topic is significant and shows great potential for scenarios with limited computational resources and real-time requirements.
2.	The experimental results are comprehensive, covering multiple benchmark datasets, and thoroughly evaluating the model's robustness and adaptability.
3.	The ability to configure the model structure at runtime makes it adaptable to different hardware constraints and application scenarios, enhancing the method's usability.

Weaknesses:

1.	The authors only compare their method between lightweight and full-scale on LLM backbones. Without the comparison with a random narrow strategy, it is hard to say whether the performance contribution is from the proposed compression method or other factors. For example, if the backbone is strong enough, a random narrow strategy may also produce similar performance.
2.	Key terminology inconsistencies. The model's name is inconsistently referred to as both "Matroyshka Re-Ranker" and "Matryoshka Re-Ranker" in the paper. It is recommended to standardize the terminology.
3.	Issues with diagram clarity. In Figure 1, the lack of directed connections in the network model creates ambiguity. If it follows a left-to-right style, the existence of same-layer connections is unclear. If it is a top-to-bottom style, the pruning results might be incorrect. For example, in the Deep-Narrow result, the width is reduced, but the depth remains unchanged.
4.	I would suggest the authors summarize their limitations and propose some future solutions to strengthen the paper.

**Questions:**

Weaknesses:

1.	The authors only compare their method between lightweight and full-scale on LLM backbones. Without the comparison with a random narrow strategy, it is hard to say whether the performance contribution is from the proposed compression method or other factors. For example, if the backbone is strong enough, a random narrow strategy may also produce similar performance.
2.	Key terminology inconsistencies. The model's name is inconsistently referred to as both "Matroyshka Re-Ranker" and "Matryoshka Re-Ranker" in the paper. It is recommended to standardize the terminology.
3.	Issues with diagram clarity. In Figure 1, the lack of directed connections in the network model creates ambiguity. If it follows a left-to-right style, the existence of same-layer connections is unclear. If it is a top-to-bottom style, the pruning results might be incorrect. For example, in the Deep-Narrow result, the width is reduced, but the depth remains unchanged.
4.	I would suggest the authors summarize their limitations and propose some future solutions to strengthen the paper.

**Reviewer Confidence:**

3: The reviewer is confident but not certain that the evaluation is correct

**Scope:**

3: The work is somewhat relevant to the Web and to the track, and is of narrow interest to a sub-community

---

### Official Review · Reviewer_Lqrv · 2024-12-03

**Novelty:** 5
**Technical Quality:** 4

**Review:**

Strengths:
- The Matryoshka Re-Ranker introduces an innovative and flexible architecture that allows for the customization of model layers and sequence lengths according to user configurations. This flexibility effectively addresses the practical challenges of deploying large language models in diverse real-world contexts.
- The paper addresses the critical challenge of balancing flexibility with the potential precision loss in LLM-based re-ranking, presenting thoughtful solutions to maintain performance.
- Extensive experiments are conducted using a wide range of datasets, including MSMARCO and the BEIR benchmarks, showcasing the thorough evaluation approach taken by the authors.

Weaknesses:
- The experimental design requires further refinement.
- There is limited comparison and justification of the core claims. Specifically, the paper lacks a comprehensive comparison of retrieval metrics and latency trade-offs across different models, presenting only efficiency comparisons among the variants of the proposed method.
- The retrieval performance of the base model is not reported.
- It is recommended to include online testing. Currently, the paper does not clarify the impact of the trade-off between reduced retrieval metrics and resource savings on actual retrieval tasks and downstream applications.

**Questions:**

Please refer to the above comments.

**Reviewer Confidence:**

3: The reviewer is confident but not certain that the evaluation is correct

**Scope:**

3: The work is somewhat relevant to the Web and to the track, and is of narrow interest to a sub-community

---

### Official Review · Reviewer_ffcp · 2024-12-03

**Novelty:** 5
**Technical Quality:** 5

**Review:**

The authors propose a flexible architecture designed to facilitate runtime customization of model layers and sequence lengths at each layer based on users’ configurations. The increased flexibility may come at the cost of precision loss. The authors handle this problem by introducing a suite of techniques to optimize the performance: 1) cascaded self-distillation, where each sub-architecture learns to preserve a precise re-ranking, 2) factorized compensation mechanism, where two collaborative Low-Rank Adaptation modules, vertical and horizontal, are jointly employed to compensate for the precision loss resulted from arbitrary combinations of layer and sequence compression. Experiments on MSMARCO (passage and documents) plus BEIR show that the proposed approach improves existing methods while effectively preserving its superior performance across various forms of compression and different application scenarios.

The idea presented in the paper is interesting, novel, and it perfectly fits the WWW 2025 call for full papers. To the best of my knowledge, this is the first contribution proposing a configurable architecture for re-ranking based on LLMs. The paper is well written. However, several issues prevent its acceptance at WWW 2025. Specifically,

1) The improvements reported in Table 1 are not complemented with statistical significance analysis. Please add statistical significance with multiple comparison correction (if needed) to provide a more comprehensive and detailed analysis of the results. Same for Table 2, although here, the differences are bigger.

2) Although the analysis regarding the efficiency of the inference reported in Figure 4 clearly shows savings of FLOP, there is no 100% clear evidence of the effect on the end-to-end latency of the query processor employing it. I would suggest that the authors complement their analysis with a more detailed evaluation of the impact of their proposal on the end-to-end retrieval time.

Overall, I feel the paper is not ready to be published at WWW 2025. This is because the experimentation lacks some complementary analysis which would make the submission much more stronger in terms of results presentation and analysis.

**Questions:**

1) Could you please comment on statistical significance of the results reported in Table 1,2?
2) Any evidence of the impact of your technique on the end-to-end latency of the query processor?

**Reviewer Confidence:**

3: The reviewer is confident but not certain that the evaluation is correct

**Scope:**

4: The work is relevant to the Web and to the track, and is of broad interest to the community

---

### Official Review · Reviewer_w3nx · 2024-12-09

**Novelty:** 5
**Technical Quality:** 4

**Review:**

This paper proposes an idea of Matroyshka re-ranker, which uses cascaded knowledge filtering and LORA compensations to facilitate runtime customization of layers and sequence lengths of LLM for re-ranking.
Factorized compensation sounds a smart idea to compensate the loss both vertically and horizontally. Extensive experiments demonstrate the effectiveness of the proposed flexible LLM-based matroyshka re-ranker. Some minor concerns include: 1) Theorem 3.1 seems an overly strong claim to be always true since both teachers and students are only sub-structures of the original LLM; 2) two lists of LoRA compensations sound expensive to tune; 3) If I remember correctly, MiniLM-based cross-encoder achieves 0.74+ on NDCG@10 on DL19 dataset, which is quite close to the number showed in Table 1, yet with a size as small as 120M in total. This raises the question of necessity of using LLM and multiple LoRAs as the re-ranker solution; 4) It could be interesting to compare Matroyshka against some traditional cross-encoders.

**Questions:**

N/A

**Reviewer Confidence:**

3: The reviewer is confident but not certain that the evaluation is correct

**Scope:**

4: The work is relevant to the Web and to the track, and is of broad interest to the community